# Current Outline of Exon Skipping Trials in Duchenne Muscular Dystrophy

**DOI:** 10.3390/genes13071241

**Published:** 2022-07-14

**Authors:** Gökçe Eser, Haluk Topaloğlu

**Affiliations:** Department of Pediatrics, Yeditepe University School of Medicine, İstanbul 34755, Turkey; drgokceeser@gmail.com

**Keywords:** DMD, therapy, exon skipping, antisense oligonucleotide, clinical trials

## Abstract

Molecular treatments for Duchenne muscular dystrophy (DMD) are already in clinical practice. One particular means is exon skipping, an approach which has more than 15 years of background. There are several promising clinical trials based on earlier works. The aim is to be able to initiate the production of enough dystrophin to change the rate of progression and create a clinical shift towards the better. Some of these molecules already have received at least conditional approval by health authorities; however, we still need new accumulating data.

## 1. Introduction

Duchenne muscular dystrophy (DMD) is a life-threatening neuromuscular disease that results from mutations in the dystrophin gene on the X chromosome, leading to severe and progressive muscle destruction. According to a meta-analysis in 2020, the global prevalence of DMD was estimated at 7.1 cases per 100,000 men and 2.8 cases per 100,000 people in the general population [1]. DMD was described in 1868 by the French neurologist Guillaume-Benjamin Duchenne, who studied muscle biopsy samples from patients [2]. In the 1980s, the structure of the dystrophin gene and protein was revealed, and with these developments, studies on DMD treatment gained momentum [3]. Research has moved into new treatment strategies, clinical trials, and drug approvals.

The *DMD* gene is one of the largest known genes in humans, representing about 0.1% of the entire genome [4]. As a result of mutations in the dystrophin gene that disrupt the reading frame, the dystrophin protein, which ensures the endurance of the muscles, cannot be produced, or is produced incorrectly (Figure 1). The reading frame is a nucleotide triplet sequence potentially translatable into a polypeptide and is decided by the positioning of a codon that begins translation. The reading frame rule characterized by out-of-frame (inactivating) DMD gene mutations in severe DMD and in-frame (residual function) DMD gene mutations in BMD is linked with protein results in around 75% of cases.

Dystrophin stabilizes skeletal muscle fibers by binding F-actin to the extracellular matrix [5]. Dystrophin is a 427 kilodalton protein localized below the sarcolemma and constitutes 5% of the sarcolemmal cytoskeletal proteins and 0.1% of all muscle protein [6]. It is predominantly found in the heart and smooth and skeletal muscle. Interestingly, splice variants of dystrophin are also expressed in the brain, retina, and Schwann cells [7]. There is also no reference standard for the level of dystrophin in DMD patients. Studies on patient cohorts with Becker muscular dystrophy and X-linked cardiomyopathy, milder forms of muscular dystrophy, show that levels of dystrophin below 30% in the skeletal muscle are adequate to prevent the start of symptoms [8]. The average dystrophin level in DMD patients’ muscle samples is 1.3%, ranging from 0.7% to 7% of the healthy average. In comparison, it is higher in BMD muscle samples (ranging from 10% to 90%), with an average dystrophin level of 33% [9].

The dystrophin protein consists of four domains. The amino-terminal domain has homology with α-actinin and contains between 232 and 240 amino acid residues. Similar to its spectrin, the central domain consists of 25 triple repeats and contains about 3000 amino acid residues. There is also a cysteine-rich domain consisting of 280 amino acids. The carboxy-terminal domain contains 420 protein residues [10]. These four domains interact with α-dystroglycan extracellularly, dystroglycan, sarcoglycan, and sarcospan in the membranes, and dystrobrevin, syntrophin, and nNOS in the cytoplasm to form the dystrophin complex [4]. When dystrophin protein translation is prematurely interrupted, its critical function is lost. Dystrophin will become dysfunctional due to mutations that affect the essential actin-binding domain or the binding domain for β-dystroglycan, and patients will exhibit the DMD phenotype. Furthermore, a DMD phenotype often develops when the central domain has a large deletion (>36 exons) [11]. Frame-shift mutations often cause a DMD phenotype, although mutations before exon 8 can cause BMD instead due to the different start locations found in exon 8. Additionally, exon 44 frame-shift mutations typically promote a milder disease than the typical form of DMD [12].

Exon skipping modifies the splicing process to make an out-of-frame mutation into an in-frame mutation. As a result, this treatment changes a DMD boy into a less severe in-frame BMD. However, mutations that preserve the reading frame allow for the production of partially functional dystrophin [13]. This dystrophin is found in Becker muscular dystrophy (BMD), which is milder than DMD and is associated with a later onset of symptoms and slower disease progression in BMD [14].

DMD has historically been diagnosed using muscle biopsy, histology, and specialized pathology methods such as immunohistochemistry or immunoblotting. Nevertheless, new developments in molecular genetic testing, particularly the next-generation sequencing technologies, have drastically changed how DMD is diagnosed.

The primary laboratory test for DMD is creatine kinase (CK) serological levels. The DMD myonecrosis and associated inflammation can be monitored using the CK level. The most popular technique for confirming DMD is genetic testing. Since roughly 70% of DMD patients have a single-exon or multi-exon deletion or duplication in the dystrophin gene, multiplex ligation-dependent probe amplification (MLPA) or comparative genomic hybridization array are typically performed as the initial confirming test. Next-generation sequencing performed at the genomic and complementary DNA (cDNA) level can identify point mutations (nonsense or missense), small deletions, duplications, and insertions, among other mutations. When no mutations are detected using MLPA, array, or Sanger sequencing, muscle biopsy can be performed to determine whether dystrophin is appropriately localized, absent/reduced, or has a changed size utilizing Western blotting and immunofluorescence analysis [15]. Several proteomic and metabolomic investigations conducted on both human and animal models have found potential DMD biomarkers. Numerous studies have discovered differences in serum protein profiles in DMD patients by using various methods to find potential biomarkers and profile serum proteins, such as stable isotope labeling, mass spectrometry (MS)-based proteome screening, multiplexed antibody, or aptamer-based assays [16].

Therapeutic approaches aimed at partially restoring functional muscular dystrophin and reducing subsequent downstream pathogenic mechanisms in patients with DMD have been developed. Exon skipping, read-through nonsense mutations, DMD gene modification, delivery of micro- or full-length dystrophin genes, progenitor cell regulation, gene replacement, and skeletal muscle cell transplantation are only a few of the molecular biology strategies available for the therapy of DMD. Exon skipping therapy is one of the promising developments for DMD and is well suited to all deletion mutations that may be rectified by inducing favorable exon skipping, enabling to reframe the transcript and restore the dystrophin protein translation, resulting in the production of a BMD-like molecule. In this review, we discuss exon skipping therapy and clinical studies that have been developed for 20 years.

## 2. Pathogenesis of DMD

The *dystrophin* gene is the largest gene identified in humans, with a length of approximately 2.3 megabases, located in the Xp21.2 region on the short arm of the X chromosome [17]. Introns make up 99% of the gene and the coding sequence is 86 exons. However, since the 7 promoter binds to the first exon, 79 exons are functional. *The dystrophin gene* has multiple gene promoters that direct the expression of different mRNA isoforms. The “full-length” 14 kb mRNA using the three gene promoters (Dp427B, Dp427M, Dp427P) contains all 79 functional exons and encodes the 427 kDa membrane cytoskeleton protein (dystrophin). These three isoforms are produced from promoter regions that have the same number of exons but are independent of each other in the brain, muscle, and cerebellar Purkinje Neurons [18]. The DMD gene also contains multiple downstream distal promoters encoding shorter mRNA and protein isoforms called Dp260, Dp140, and Dp116, and the shortest, Dp71, shows relatively ubiquitous expression in non-muscle and nerve cells. The many isoforms produced by the dystrophin gene through alternative splicing events create greater protein diversity and regulate the complex expression of dystrophin [19].

The muscle dystrophin protein (Dp427m), a 427 kDa protein, is the most important and relevant in DMD. The four-domain dystrophin protein attaches tightly to the glycoprotein complex, stabilizing the sarcolemma. In this way, degradation of the glycoprotein complex by proteases is prevented. The absence of dystrophin under the sarcolemma makes the muscle fibers more vulnerable during muscle contraction. Β-dystroglycan (β-DG), one of the essential transmembrane components, is reduced due to the dystrophin–glycoprotein complex disassembling due to the absence of functioning dystrophin. β-DG acts as a bridge between extracellular laminin and dystrophin, guaranteeing sarcolemmal stability. Additionally, β-DG participates in signaling through interactions with certain intracellular partners controlled by post-translational alterations such as tyrosine residue phosphorylation [20]. Muscle fibers with incomplete functional dystrophin are easily damaged, eventually replacing muscle tissue with fibrotic and adipose tissue and causing progressive loss of muscle function [21].

## 3. Exon Skipping

The dystrophin gene has a high mutation rate compared to other human genes, with around one-third of changes arising de novo. Large deletions (68%), duplications (10%), and small variants (nonsense, missense, deletion, insertion, and splicing variants) (22%) are the most common types of mutations in the DMD gene. Less than 1% of all mutations are atypical, including deep intronic variations and complex intragenic rearrangements (Table 1) [22].

Through frameshifts or nonsense mutations, the production of the functional dystrophin protein is abolished. Deletions and duplications can occur in any region of the dystrophin gene, but it is known that these mutations occur more frequently in two regions. These are the central region and the 5′ end of the gene. The central region of the gene is the most frequently mutated and contains exons 45–55, while the 5′ end region contains exons 2–19 with genetic break points. The introns of the gene are in exons 2 and 7 [23]. There is no simple relationship between the size of the deletion and the resulting clinical disease. Effects on phenotype depend on whether it distorts the reading frame. In patients with DMD, deletions and duplications distort the reading frame, resulting in unstable RNA production. For this reason, dystrophin protein concentrations are reduced to an almost undetectable level [24]. The DMD gene’s 79 exons frequently start and end with blunt ends, with each exon’s triplet codon completely encoding the amino acids that it codes for. In particular, in-frame deletions may allow the production of partially functional dystrophin proteins [25]. This partial functional dystrophin is associated with BMD, a less severe disease. Exon skipping therapy targets the difference between DMD and BMD mutations. The goal is to restore the reading frame to allow DMD patients to make a BMD-like protein. Antisense oligonucleotides (AONs) have been developed to achieve exon skipping. They are small pieces of modified DNA or RNA. Their job is to specifically hybridize to a target exon and hide that exon from the splicing machine [26]. AONs can alter mRNA expression through various mechanisms, including ribonuclease H-mediated degradation of mRNA, direct steric blocking, and modulation of exon content via splice site binding onto pre-mRNA. Before splicing, oligonucleotide drugs bind to the dystrophin pre-mRNA so that the corresponding exon is skipped when reading the mature mRNA while producing the protein. This mechanism restores the corrupted reading frame and allows the production of an internally deleted but largely functional protein [27].

ASOs undergo different chemical modifications to improve their metabolic stability, binding specificity and affinity. The main change is the sugar modification. 2′-O-methyl phosphorothioate (2OMePS) AONs and phosphorodiamidate morpholino oligomers (PMOs) have been the preferred therapeutics for exon skipping in DMD [28]. ASOs also mediate the regulation of microRNA, mRNA polyadenylation signals, and binding between proteins and pathogenic RNA, which are involved in many diseases [27,28].

Exon skipping is a mutation-specific approach. Which exon is skipped depends on the size and location of the mutation. For example, Casimersen, an exon 45 skipping molecule, is an intravenous infusion drug that binds to exon 45 of the *dystrophin* gene pre-mRNA and causes this exon to be skipped during mRNA processing. This allows patients to produce an internally shortened but functional dystrophin protein [29]. Since the deletions are more common around exons 45 and 55, studies on skipping exons in this area involve larger patient groups. Out of total patients, exon 51 skipping can be used in 14%, exon 53 skipping applies to 10%, and exon 45 skipping applies to 9%, respectively [30].

The skipping of two or more exons is required In some cases. Exon skipping is difficult even for multiple exon duplications because the effect will be muted; skipping a duplicated exon frequently restores the reading frame, whereas skipping an original exon frequently does not (Table 2).

### 3.1. Exon 51 Skipping

The first exon skipping clinical studies for DMD started in 2006 and were intended to evaluate the inclusion of the 2OMePS chemical targeting exon 51 skipping. However, there was a lack of solid evidence for the effectiveness of this medication, drisapersen, and there were safety worries about severe injection site responses that persisted after stopping the medication.

#### 3.1.1. Eteplirsen

Eteplirsen was first tested in seven patients by intramuscular injection into the extensor digitorum brevis muscle of the foot. Local injections of 0.09 mg (n:2) and 0.9 mg (n:5) eteplirsen were administered. It resulted in restoration of de novo dystrophin in the high dose group (NCT00159250) [31]. A dose-finding study was conducted in which 19 patients were treated with intravenous infusions of 0.5, 1, 2, 4, 10, and 20 mg/kg for 12 weeks. In the highest dose group of muscle biopsies, Western blot measurements showed an average increase of 4% in dystrophin levels (NCT00844597). No dose-related effects were observed in motor function tests [32]. The next placebo-controlled study was conducted on 12 patients. Four patients were divided into placebo, 30, or 50 mg/kg dose groups and followed for 24 weeks [33]. The primary outcome was dystrophin-positive myofibers measured by immunohistochemistry (NCT01396239), followed by an open-label continuation study (NCT01540409). Dystrophin-positive fibers were detected in biopsies analyzed after 24 and 48 weeks of treatment, and patients in the drug group evaluated with 6MWT showed a slower rate of decline in ambulation [34,35]. In September 2016, the FDA granted accelerated marketing authorization for eteplirsen. The phase 2 study (NCT03218995) evaluating the safety, efficacy, and tolerability of eteplirsen in 15 patients aged 6–48 months ended in March 2021. According to the published results, bronchiolitis was observed as a serious side effect in one patient (accessed on 9 December 2021: https://clinicaltrials.gov/ct2/show/NCT03218995?term=NCT03218995&draw=2&rank=1). There is an ongoing phase 3 randomized, double-blind, dose-finding, and comparison study on the safety and efficacy of high-dose eteplirsen (NCT03992430). To improve the molecule’s delivery to cells, Sarepta is conducting a two-part study of dose-determination safety trials with SRP-5051 which contains eteplirsen conjugated to an arginine-rich peptide (NCT04004065).

#### 3.1.2. Drisapersen-Suvodirsen

Drisapersen and suvodirsen developed for exon 51 have also been evaluated in clinical trials, but clinical development of these compounds has been discontinued. The drisapersen study failed to demonstrate a statistically or clinically significant improvement of the 6MWT, and the suvodirsen study results showed no change from baseline in dystrophin expression, as measured by Western blot.

### 3.2. Exon 53 Skipping

Two exon 53 skipping PMOs are available. Golodirsen (Sarepta Therapeutics) and Viltolarsen (Nippon Shinyaku).

#### 3.2.1. Golodirsen

Golodirsen was first tested in a 24-week placebo-controlled, dose-escalation study of 12 participants [36]. Thirty-nine participants were then enrolled in a long-term extension study (NCT02310906). Forty-eight weeks later, dystrophin immunoblot analysis showed a drug-induced increase in dystrophin of 1% [37]. Golodirsen was approved in the United States in December 2019 [38].

#### 3.2.2. Viltolarsen

Viltolarsen, the drug targeting the other exon 53, received accelerated FDA approval for DMD patients in August 2020 [39]. An exploratory study was conducted with 10 participants in 2013 (NCT02081625). Viltolarsen demonstrated in a phase 1/2 study that intravenous infusions of 80 mg/kg/week for 24 weeks in Japanese Duchenne muscular dystrophy patients increased mean dystrophin levels by 2.8% by Western blot analysis [40]. Another phase 2 study (NCT02740972) evaluating the safety and intravenous dose of viltolarsen in DMD boys aged 4–9 years showed significant drug-induced dystrophin production in both viltolarsen dose cohorts (mean [range]: 5.7% [3.2–10.3] in the 40 mg/kg group; 5.9% in the 80 mg/kg group) [41]. Another phase 2 study of viltolarsen concluded in November 2021. According to the results, there was no decline in functional tests over 2 years, whereas there was a significant decline in the historical control group [42]. Placebo-controlled trials are ongoing for golodirsen and viltolarsen (NCT02500381 and NCT04060199-NCT04956289). Moreover, the codes have not been broken yet. This molecule has the approval of the Ministry of Health, Labor and Welfare of Japan.

### 3.3. Exon 45 Skipping

#### Casimersen

Casimersen received accelerated approval from the FDA in February 2021 for treating DMD patients eligible for exon 45 skipping. After 48 weeks of treatment, the mean dystrophin protein level was 1.736% in the treatment group compared to 0.925% in the placebo group (NCT02500381) [43].

## 4. Conclusions

Exon skipping therapy with antisense oligonucleotides is one of the revolutionary developments for DMD disease. Currently, there are four exon skipping drugs conditionally approved for DMD (Table 3).

This treatment brings some difficulties with it. First, the distribution of ASOs to muscle cells and other cells varies. On the other hand, only deficient levels of dystrophin are restored by AONs. Whether these amounts are enough to slow down the disease’s progression is the question. Even though there will be individual patient variations, we think that restoring even small levels of dystrophin may somewhat slow the disease’s course.

Eteplirsen, golodirsen, and viltolarsen should all be recognized as having been authorized based on dystrophin levels, not on their functional effects on muscles. It is essential to monitor renal function given the results of nephrotoxicity encountered with other ASOs.

The FDA has reportedly put a clinical hold on testing for Sarepta Therapeutics’ experimental compound SRP-5051. The choice was based on a patient in part B of the phase 2 MOMENTUM study (NCT04004065) who experienced a major adverse event of low magnesium following treatment with high-dose SRP-5051.

A very small fraction of PMO drugs given by intravenous infusion reaches the myofibril core [44]. Several strategies are being used to improve the effectiveness of AON delivery to the heart and muscle and raise levels of dystrophin repair. The problem is that many of these methods may not be safe for people and have not yet been evaluated on humans.

By masking crucial splicing sites, specific exon skipping can be accomplished using synthetic oligonucleotides or viral vectors expressing modified small nuclear RNAs (snRNAs). A “double-target” U7 Sm OPT was used to skip exon 23 of the mdx dystrophin mRNA in one of the initial experiments on the rescue of DMD by exon skipping mediated by U7 snRNA. The AAV vector was used to deliver the engineered U7 snRNA to the skeletal muscles of the mdx mice [45]. This design enabled effective exon 23 skipping, saving dystrophin. However, it was shown that the therapeutic benefit of exon skipping diminished over time, as the AAV vector genome was lost from muscle fibers [46].

The use of the utrophin gene as therapy for the treatment of DMD in animal models is discussed in another study. Miniaturized utrophin, a very effective and non-immunogenic replacement for dystrophin, is employed in this instance to stop the most harmful histological and physiological effects of DMD in small and large animal models [47].

Instead of using the full-length normal protein, all treatment attempts to restore dystrophin in the muscle of DMD patients use semi-functional, biochemically abnormal variants of dystrophin. We cannot anticipate that the administration of partially functioning dystrophin would cure skeletal muscle tissue and restore it to normal muscle; success is defined as a Becker-type muscle and phenotype.

The current exon skipping therapies have the potential to benefit 30% of patients; however, the findings on these treatments are mainly focused on dystrophin restoration at low levels (less than 6%), and additional research is needed to determine long-term clinical outcomes. In this area, there is still a great deal of space for development.

In addition, the necessity of administering intravenous doses every week imposes a burden on families and patients. However, this problem can be prevented with the home treatment application. Genome editing has a more long-lasting impact, but this treatment uses substantial dosages of viral vectors. Unfortunately, therapy with large dosages of viral particles resulted in three deaths in individuals with myotubular myopathy due to immunological reactions. Patients who do not already have an immunological reaction to the particular AAV serotype are also suitable for gene therapy [48]. The immune response to dystrophin and the AAV vector used to deliver the micro-gene is frequently linked to dystrophin replacement. Immunomodulatory medications such as rituximab and VBP6 may abolish the cellular immunological response to micro-dystrophin and only partially inhibit the interferon-γ response [49].

Therapies for reading frame restoration rely on the expressed *dystrophin* gene, which only happens in skeletal and cardiac muscle and not in adipose or fibrotic tissues. As a result, over time, muscle loss will cause the number of target transcripts to decrease. Earlier treatment before connective tissue enlargement may inevitably show a more significant clinical benefit relative to the disease course [3]. At this stage, we do not entirely know the effect of exon skipping therapies on heart function, which can potentially be a compromise. As it stands, exon skipping is a feasible method for treatment provided that deletions match. At this stage, we do not know whether it may in future be combined with other genetic methods such as gene therapy. There is hope that these molecules can modify the evolution of DMD.

## Figures and Tables

**Figure 1 genes-13-01241-f001:**
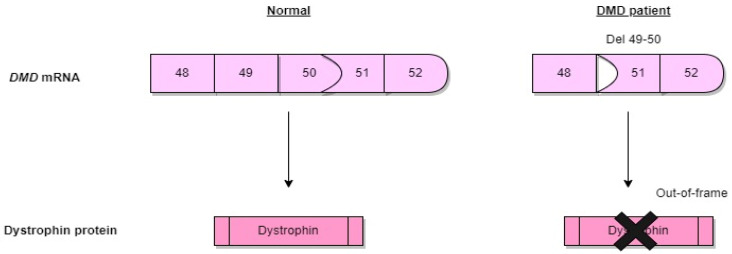
The schematic diagram of the dystrophin.

**Table 1 genes-13-01241-t001:** Variants of dystrophinopathies [22].

Mutation Type	BMD	DMD
Large deletion	80–81%	61–66%
Large duplication	6–9%	11–13%
Complex rearrangements	<0.5%	<0.5%
Nonsense	3%	12–13%
Frameshift	2%	6–8%
Missense	0.5–0.7%	0.3–0.9%
Total small variants	11–13%	23–26%

**Table 2 genes-13-01241-t002:** History of active and ongoing exon skipping trials.

Code	Phase	Actual Study Start Date	Actual Study Completion Date	Patient	Study Design	Result
**Eteplirsen (exon 51 skipping)**						
NCT00159250	Phase½2	26 October 2007	31 March 2009	7	Non-randomized	dystrophin expression is increased in 44–79% of myofibrils
NCT00844597	Phase½2	January 2009	December 2010	19	Non-randomized	mean dystrophin intensity increased from 8.9% to 16.4%
NCT01396239	Phase 2	July 2011	June 2012	12	Randomized	In 30 mg/kg eteplirsen patient group, at week 24 dystrophin-positive fibers increased to 23%, at week 48 increased to 52%
NCT01540409	Phase 2	27 February 2012	16 August 2017	12	Randomized	patients in the drug group evaluated with 6 MWT showed a slower rate of decline in ambulation
NCT03218995	Phase 2	16 August 2017	10 March 2021	15	Interventional, open-label	one patient had a serious adverse event (bronchiolitis)
NCT03992430	Phase 3	13 July 2020	Recruiting	154 participants (estimated)	Randomized, double-blind, dose-finding	estimated completion date is 30 November 2024
**SRP-5051 (exon 51 skipping)**						
NCT03375255	Phase 1	5 February 2018	19 August 2019	15	Sequential assignment, open-label	no results reported
NCT04004065	Phase 2	26 June 2019	Recruiting	60 participants (estimated)	Randomized	estimated completion date is 31 August 2024
**Golodirsen (exon 53 skipping)**						
NCT02310906	Phase I/II	13 January 2015	25 March 2019	39	Randomized, double-blind, placebo-controlled	increased dystrophin protein 16.0-fold, in treatment group loss of ambulation occurred in 9% versus 26% in control group
NCT02500381	Phase 3	28 September 2016	Recruiting	222 participants (estimated)	Double-blind, placebo-controlled	estimated study completion date 30 April 2024
**Viltolarsen (exon 53 skipping)**						
NCT02740972	Phase 2	December 2016	April 2018	16	Randomized, placebo-controlled	significant dystrophin production in biceps biopsies (mean [range]: 40 mg/kg group: 5.7% [3.2–10.3]; 80 mg/kg group: 5.9% [1.1–14.4] of normal)
NCT03167255	Phase 2	6 July 2017	15 November 2021	16	Non-randomized, open-label	in the treatment group, there was no decline until the 109th week, whereas there was a functional decrease in the historical control group
NCT04060199	Phase 3	14 April 2020	Recruiting	74 participants (estimated)	Randomized, double-blind, placebo-controlled	estimated completion date is December 2024
NCT04956289	Phase 3	1 July 2021	Recruiting	22 participants (estimated)	Open-label	estimated completion date is May 2024
**Casimersen (exon 45 skipping)**						
NCT02530905	Phase 1	8 October 2015	3 October 2018	12	Randomized, double-blind, placebo-controlled	dose-escalation study followed by an open-label study
NCT02500381	Phase 1/2	28 September 2016	Recruiting	222 participants (estimated)	Double-blind, placebo-controlled	estimated completion date 30 April 2024

**Table 3 genes-13-01241-t003:** Features and properties of exon skipping drugs in DMD.

Drug	Sponsor	Dose	Route of Administration	Side Effect
Eteplirsen Exondys 51	Sarepta Therapeutics	30 mg/kg	Intravenous infusion weekly	headache, fever, falls, abdominal pain, cough, and nausea
Golodirsen SRP-4053 Vyondys 53	Sarepta Therapeutics	30 mg/kg	Intravenous infusion weekly	headache, fever, fall, cough, vomiting, abdominal pain, cold symptoms (nasopharyngitis), and nausea
Viltolarsen Viltepso	NS Pharma	80 mg/kg	Intravenous infusion weekly	upper respiratory tract infection, injection site reaction, cough, and pyrexia (fever)
Casimersen Amondys 45; SRP-4045	Sarepta Therapeutics	30 mg/kg	Intravenous infusion weekly	upper respiratory tract infection, cough, pyrexia, headache, arthralgia, and oropharyngeal pain

## Data Availability

Not applicable.

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
