# Peer review of "Current Outline of Exon Skipping Trials in Duchenne Muscular Dystrophy"

_genes, 2022, doi:10.3390/genes13071241_

Round 1

Reviewer 1 Report

The authors have summarized the effect of various drugs related to exon skipping trial in DMD. The review is incomplete and needs extensive improvement both in English language and contents. I would suggest the authors to rewrite it with headings and subheadings. Also addition of figures and extra tables summarizing the important text would be very helpful for the readers. My comments are provided below

1.      In the line 22, please add the details about dystrophin level in the muscle. It is a very large protein and expressed at very low level in the muscle. Individual fibres of the patients also express varying level of dystrophin. There is also no reference standard for the level of dystrophin in DMD patients. The dystrophin level vary amongst the unaffected individual.

2.      In line 23, please add describe the detail structure of dystrophin. The author can provide a schematic diagram of the dystrophin mentioning the mutation, deletion of the exon – particularly exon 45, 51, 53 etc.

3.      In line 29, the authors should describe about the specific mutations associated with DMD and BMD by providing a table on it or citing the previous literatures where these have been discussed in details.

4.      In line 30, the authors should discuss the sequencing method – genomic and proteomics level (by mass spec) for identifying the patients.

5.      In line 37, the authors should describe about different strategies for treating DMD and why exon skipping is better/not better compared to those strategies.

6.      In line 59, the authors should discuss the interaction between, dystrophin and beta-dystroglycan and its role in DMD patients as an example.

7.      In line, 61, the authors should provide a table about the mutations in dystrophin reported previously mentioning the deletion/duplication. How many percentage of these individual mutation contributes to the disease phenotypes?

8.      In line 74, the authors have mentioned about blunt end and they should discuss about it.

9.      In line 95, please discuss in details about exon 45 skipping strategy as an example. In line 96, please mention that out of total patients, the strategies of exon skipping 51, 53 and  45 applies to 14%, 10% and 9%, respectively. Also there are many other mutations are reported in such patients in other region of the dystrophin. Here the authors should discuss the demerits of these strategies.

10.   In table 1, the authors should add an extra column about percentage of patients these drugs are applicable.

11.   In line 100 onwards for each drugs, the authors should add a subheading about the drug. It would be better, if the author could provide a separate table for these four drugs mentioned their dose, route of administration, time period, effectiveness in patients, side effects etc.

12.   The authors should describe the side effect of these drugs in details.

13.    In conclusion, the authors should also discuss the drawback of these strategies.

Author Response

In the line 22, please add the details about dystrophin level in the muscle. It is a very large protein and expressed at very low level in the muscle. Individual fibres of the patients also express varying level of dystrophin. There is also no reference standard for the level of dystrophin in DMD patients. The dystrophin level vary amongst the unaffected individual.

We've added details about dystrophin, thank you.

In line 23, please add describe the detail structure of dystrophin. The author can provide a schematic diagram of the dystrophin mentioning the mutation, deletion of the exon – particularly exon 45, 51, 53 etc.

 We've mentioned about the structure of protein and added a schematic diagram. Thank you.

In line 29, the authors should describe about the specific mutations associated with DMD and BMD by providing a table on it or citing the previous literatures where these have been discussed in details.

We described about specific mutations and added a table. Thank you. 

In line 30, the authors should discuss the sequencing method – genomic and proteomics level (by mass spec) for identifying the patients.

We've discussed that, Thank you.

In line 37, the authors should describe about different strategies for treating DMD and why exon skipping is better/not better compared to those strategies.

We've added different treatment strategies in conclusion part, thank you.

In line 59, the authors should discuss the interaction between, dystrophin and beta-dystroglycan and its role in DMD patients as an example.

We've mentioned about the link between dystrophin and BDG, thank you.

In line, 61, the authors should provide a table about the mutations in dystrophin reported previously mentioning the deletion/duplication. How many percentage of these individual mutation contributes to the disease phenotypes?

We've added a table that you mentioned, thank you.

In line 74, the authors have mentioned about blunt end and they should discuss about it.

We've discussed more about blunt end, thank you.

In line 95, please discuss in details about exon 45 skipping strategy as an example. In line 96, please mention that out of total patients, the strategies of exon skipping 51, 53 and  45 applies to 14%, 10% and 9%, respectively. Also there are many other mutations are reported in such patients in other region of the dystrophin. Here the authors should discuss the demerits of these strategies.

We have added the sections you mentioned, thank you.

In table 1, the authors should add an extra column about percentage of patients these drugs are applicable.

We did not include this section, since not all studies have yet concluded patient recruitment and we have indicated the number of patients in studies that closed patient recruitment.

In line 100 onwards for each drugs, the authors should add a subheading about the drug. It would be better, if the author could provide a separate table for these four drugs mentioned their dose, route of administration, time period, effectiveness in patients, side effects etc.

We've added the table as you requested but, Since studies are still ongoing, we did not add the effectiveness in patients to the table in order not to contradict the information we provide, in the future. Thank you

The authors should describe the side effect of these drugs in details.

We've mentioned in details, thank you

 In conclusion, the authors should also discuss the drawback of these strategies.

We've discussed the drawback in conclusion part, thank you.

Reviewer 2 Report

Line 7. Instead of “is approaching” would say “are already in clinical practice”

Line 8. Typo: “could ve”  should be “could be”

Line 16: DMD if it is gene, should be italicized

Lines 26-27.  The concept of “reading frame” should first be explained and also emphasized that it applies only to deletion mutations. Also should explain that “in frame” “out of frame” rule applies to the majority (80-90%) of deletions but not all of them.

Line 27. “partial functional dystrophin” should read “partially functional dystrophin”

Lines 30-34: these introductory sentences should go earlier in the introduction prior to explaining the DMD gene.

Line 35-36: “Therapeutic approaches aimed at partially restoring functional muscular dystrophin in patients with DMD have been demonstrated”  This sentence needs a better verb like “have been developed”

Line 61: In the sentence “About 60% of dystrophin mutations are insertions and deletions that lead to a large frameshift mutations and about 40% are point mutations or small frameshift rearrangements. In other words, 2/3 of DMD cases represent single or multiple exon deletions”

-          What do authors mean by “insertion” mutation?  should be clarified “frameshift insertions?” or else. Also exon duplications that represent about 10% are not mentioned.

-          40% point mutations or small frameshift rearrangements: 40% seems too high and should be around 15-30%

-          2/3 represent exon deletions: add “about” or “approximately” 2/3

Line 72: The general statement of “The DMD gene can tolerate deletion of one or more exons and produce dystrophin protein with reduced functionality.” is not scientifically accurate. It depends on what exon and also depends on other factors many still unknown. Single deletions can lead to disruption of the entire protein synthesis leading to a sever phenotype.

Line 79: “Antisense oligonucleotides (AONs) developed to achieve exon-skipping.” Missing verb “have been developed”

Line 94: “Since the deletions are concentrated between exons 45 and 55”. “are concentrated ….” should be replaced with something like “are more common around exons 45 to 55”

Line 98, Table 1 for all the subheadings (above each box):

Authors should mention right after each product that which exon is supposed to be skipped, for example, Eteplirsen (exon 51 skipping) etc

Lines 121-122: reasons should be mentioned with references and the name of comoanies developing them. Authors can refer to company news releases. Also should say that drisapersen was the first one developed and was a subcutaneous injection.

Drisapersen: study failed to demonstrate a statistically or clinically significant improvement of the 6MWT.

Suvodirsen: results showed no change from baseline in dystrophin expression, as measured by western blot.

Lines 126 and 127:  sentences should not start with numbers. “thirty-nine participants” etc

Line 130: “explotary study of viltolarsen”  >> “exploratory”

Line 136: “rhase2 study”  >> “phase 2 study”

Lines 139-142: “We have hands-on experience with this molecule. The trial is placebo controlled, and the codes have not been broken yet. We rely on the good news that this molecule can lead to dytrophin restoration up to 6% by immunoblot”

“We have hands-on experience with this molecule.” Should be removed

It is not scientific to say “We rely on the good news that this molecule can lead to dytrophin restoration up to 6% by immunoblot”    How do you know before they disclose the data? And remove “rely on the good news etc”

Line 150:  put at least a few references for “very small fraction of PMO drugs given by intravenous infusion reach the myofibril core.

Line 151-152: put a few references for “conjugates and muscle-directed peptides are added to drugs for facilitating the 152 penetration of ASOs  There are now multiple company for such products and the authors have to reference to their work and elaborate on this different approach of ASO delivery. These include ab’s against transferrin receptor with efforts by Dyne and Avidity companies.

-          Also should mention that there is, at least theoretically, potential immune response development against such conjugates/antibodies. This needs further data gathering by such companies as they progress in their product development.

Lines 154-155:  Please provide references for “Earlier treatment at younger ages before connective tissue is formed is expected to show greater clinical benefit relative to the disease course

Line 157: “By all means these molecules hopefully modify the evolution of DMDshould be re-written with removal of “by all means” and “hopefully”. Like “there is hope… etc”

146 Conclusion section:

Discussion is short and other issues need to be elaborated more WITH REFERENCES.

-          Exon skipping using U7 snRNA should be discussed (such as discussed in Goyenvalle et al. Mol Ther 2009)

-          Should mention that current exon 45, 51, and 53 skips could potentially help about 30% of patients but the data on these therapies are mainly on dystrophin restoration at low levels (less than 6%) and long term clinical outcome measures need to be investigated further and studies are ongoing.

-          Should be emphasized that there is a lot of room for improvement still in this domain

-          The risk of hypomagnesemia with some of the AON’s should be mentioned with reference.

-          In the discussion, should mention that exon skipping is one of the strategies and in the future there is possibility of it being used in conjunction with other genetic approaches such as gene therapy.

-          There was no discussion about the AAV safety issues. This is very important and authors should incorporate that in their paper with some references.

Author Response

Line 7. Instead of “is approaching” would say “are already in clinical practice”

Yes, I have done that, thank you. 

Line 8. Typo: “could ve”  should be “could be”

Yes, we changed that, thank you.

Line 16: DMD if it is gene, should be 

We italicized it. Thank you

Lines 26-27.  The concept of “reading frame” should first be explained and also emphasized that it applies only to deletion mutations. Also should explain that “in frame” “out of frame” rule applies to the majority (80-90%) of deletions but not all of them.

We explained reading frame rule (including in-frame/out-of-frame rules) Thank you

Line 27. “partial functional dystrophin” should read “partially functional dystrophin”

We have done that, thank you

Lines 30-34: these introductory sentences should go earlier in the introduction prior to explaining the DMD gene.

We've changed that, thank you.

Line 35-36: “Therapeutic approaches aimed at partially restoring functional muscular dystrophin in patients with DMD have been demonstrated”  This sentence needs a better verb like “have been developed”

We've changed the verb, thank you.

Line 61: In the sentence “About 60% of dystrophin mutations are insertions and deletions that lead to a large frameshift mutations and about 40% are point mutations or small frameshift rearrangements. In other words, 2/3 of DMD cases represent single or multiple exon deletions”

-          What do authors mean by “insertion” mutation?  should be clarified “frameshift insertions?” or else. Also exon duplications that represent about 10% are not mentioned.

-          40% point mutations or small frameshift rearrangements: 40% seems too high and should be around 15-30%

-          2/3 represent exon deletions: add “about” or “approximately” 2/3

We made the changes you mentioned and clarified the insertion. Thank you.

Line 72: The general statement of “The DMD gene can tolerate deletion of one or more exons and produce dystrophin protein with reduced functionality.” is not scientifically accurate. It depends on what exon and also depends on other factors many still unknown. Single deletions can lead to disruption of the entire protein synthesis leading to a sever phenotype.

We removed that part, thank you.

Line 79: “Antisense oligonucleotides (AONs) developed to achieve exon-skipping.” Missing verb “have been developed”

Yes, we've done that, thank you.

Line 94: “Since the deletions are concentrated between exons 45 and 55”. “are concentrated ….” should be replaced with something like “are more common around exons 45 to 55”

Yes, we've done that, thank you. 

Line 98, Table 1 for all the subheadings (above each box):

Authors should mention right after each product that which exon is supposed to be skipped, for example, Eteplirsen (exon 51 skipping) etc

Yes, we've added those, thank you. 

Lines 121-122: reasons should be mentioned with references and the name of comoanies developing them. Authors can refer to company news releases. Also should say that drisapersen was the first one developed and was a subcutaneous injection.

At this stage we prefer not to refer to company news as bias happens.

Drisapersen: study failed to demonstrate a statistically or clinically significant improvement of the 6MWT.

Added

Suvodirsen: results showed no change from baseline in dystrophin expression, as measured by western blot.

Added

Lines 126 and 127:  sentences should not start with numbers. “thirty-nine participants” etc

Corrected

Line 130: “explotary study of viltolarsen”  >> “exploratory”

Corrected

Line 136: “rhase2 study”  >> “phase 2 study”

Corrected

Lines 139-142: “We have hands-on experience with this molecule. The trial is placebo controlled, and the codes have not been broken yet. We rely on the good news that this molecule can lead to dytrophin restoration up to 6% by immunoblot”

“We have hands-on experience with this molecule.” Should be removed

It is not scientific to say “We rely on the good news that this molecule can lead to dytrophin restoration up to 6% by immunoblot”    How do you know before they disclose the data? And remove “rely on the good news etc”

The parts that mentioned were removed, thank you.

Line 150:  put at least a few references for “very small fraction of PMO drugs given by intravenous infusion reach the myofibril core.”

Proper reference added, thank you.

Line 151-152: put a few references for “conjugates and muscle-directed peptides are added to drugs for facilitating the 152 penetration of ASOs”  There are now multiple company for such products and the authors have to reference to their work and elaborate on this different approach of ASO delivery. These include ab’s against transferrin receptor with efforts by Dyne and Avidity companies.

 Because ours is a simple clinical review of exon skipping, we have preffered not to include this into the text as such developments are still a bit far from clinical trials. I hope it will be considered by the referee. Thank you. 

-          Also should mention that there is, at least theoretically, potential immune response development against such conjugates/antibodies. This needs further data gathering by such companies as they progress in their product development.

 We have briefly mentioned this baaed on the fact that immunology is essentially compromised in gene therapy trials. Exon skipping however may be safer in this respect. 

Lines 154-155:  Please provide references for “Earlier treatment at younger ages before connective tissue is formed is expected to show greater clinical benefit relative to the disease course”

 Added.

Line 157: “By all means these molecules hopefully modify the evolution of DMD” should be re-written with removal of “by all means” and “hopefully”. Like “there is hope… etc”

 Changed, thank you.

146 Conclusion section:

Discussion is short and other issues need to be elaborated more WITH REFERENCES.

-          Exon skipping using U7 snRNA should be discussed (such as discussed in Goyenvalle et al. Mol Ther 2009)

-          Should mention that current exon 45, 51, and 53 skips could potentially help about 30% of patients but the data on these therapies are mainly on dystrophin restoration at low levels (less than 6%) and long term clinical outcome measures need to be investigated further and studies are ongoing.

-          Should be emphasized that there is a lot of room for improvement still in this domain

-          The risk of hypomagnesemia with some of the AON’s should be mentioned with reference.

-          In the discussion, should mention that exon skipping is one of the strategies and in the future there is possibility of it being used in conjunction with other genetic approaches such as gene therapy.

  •          There was no discussion about the AAV safety issues. This is very important and authors should incorporate that in their paper with some references.

We've added the parts you requested and extended the discussion section in the conclusion part. Thank you.

Round 2

Reviewer 1 Report

The authors have addressed all my concerns in the revised manuscript. The quality of the revised manuscript is now improved. I support the publication of the revised manuscript.

Reviewer 2 Report

Thank you for making the corrections/improvements.